# Chronically implanted Neuropixels probes enable high-yield recordings in freely moving mice

Ashley L Juavinett[1], George Bekheet[2], Anne K Churchland[3]*

[1]Division of Biological Sciences, University of California, San Diego, San Diego, United States; [2]University of Connecticut School of Medicine, Farmington, United States; [3]Cold Spring Harbor Laboratory, Cold Spring Harbor, United States

**Abstract** The advent of high-yield electrophysiology using Neuropixels probes is now enabling researchers to simultaneously record hundreds of neurons with remarkably high signal to noise. However, these probes have not been well-suited to use in freely moving mice. It is critical to study neural activity in unrestricted animals for many reasons, such as leveraging ethological approaches to study neural circuits. We designed and implemented a novel device that allows Neuropixels probes to be customized for chronically implanted experiments in freely moving mice. We demonstrate the ease and utility of this approach in recording hundreds of neurons during an ethological behavior across weeks of experiments. We provide the technical drawings and procedures for other researchers to do the same. Importantly, our approach enables researchers to explant and reuse these valuable probes, a transformative step which has not been established for recordings with any type of chronically-implanted probe.
DOI: https://doi.org/10.7554/eLife.47188.001

## Introduction

Observing behavior and recording neural activity in freely moving animals is crucial for our understanding of how the brain operates. Electrophysiology in freely moving rodents has been used to observe place and grid cell dynamics (*Hafting et al., 2005*; *O'Keefe and Dostrovsky, 1971*), cortical dynamics during attentional control (*Bolkan et al., 2017*), the role of oscillations during fear learning (*Stujenske et al., 2014*), whisking behavior during exploration (*Kerekes et al., 2017*), the effect of environmental context on neural activity (*Whitlock et al., 2012*), and the control of sensory selection in divided attention (*Wimmer et al., 2015*), to name a few. Although freely moving recordings can be challenging, recording from unrestrained mice enables researchers to investigate behaviors that involve natural movements and offers ethologically valid insight into neural activity (*Juavinett et al., 2018*; *Markowitz et al., 2018*). Electrophysiology in freely moving animals is commonly performed with static electrode arrays or microdrives (*Okun et al., 2016*; *Vandecasteele et al., 2012*; *Voigts et al., 2013*). These techniques have contributed much to the field, but are not at pace with the spatiotemporal coverage of cutting edge recording techniques, such as Neuropixels probes (*Jun et al., 2017*; *Steinmetz et al., 2018*). Given the experimental tractability of the mouse and the increasing interest in ethological approaches in neuroscience research, we sought to develop a system that would enable researchers to perform repeatable high-yield recordings.

Recent advancements in semiconductor technology have enabled the development of high-density silicon probes known as Neuropixels (*Jun et al., 2017*). The linear recording shank can record from 384 contacts across 3.84 mm (selectable from 960 available sites on a 10 mm length shank). In the mouse brain, which is at most 6 mm deep, this span of contacts means researchers can

**\*For correspondence:**
churchland@cshl.edu

**Competing interests:** The authors declare that no competing interests exist.

simultaneously record from more than half of the depth of the brain. Further, Neuropixels probes have low baseline noise levels (<6 µV RMS), comparable to other silicon probes (*Steinmetz et al., 2018*). However, Neuropixels probes also have on-site amplification and digitization, thereby enabling simultaneous recording of hundreds of cells across brain regions in an unprecedented low-noise, high-throughput manner. Importantly, methods have also been developed to automatically sort spikes from these recordings, and even correct for probe drift (*Jaeyoon et al., 2017*; *Pachitariu et al., 2016*).

Neuropixels probes have already proved invaluable for neuroscientists conducting acute experiments in mice, or chronic experiments in freely moving rats (*Jaeyoon et al., 2017*; *Krupic et al., 2018*; *Vélez-Fort et al., 2018*). However, there is limited work with these probes in unrestrained mice (*Evans et al., 2018*), likely because of the difficulty designing small, lightweight recording devices. Still, there is plentiful interest in behaviors and computations that involve movements of the animal's head in space (*Vélez-Fort et al., 2018*), foraging (*Lottem et al., 2018*), pup retrieval (*Marlin et al., 2015*), or naturalistic fear responses (*Evans et al., 2018*). Further, although these probes have been very successful in freely moving rats (*Jaeyoon et al., 2017*; *Krupic et al., 2018*), there is not an established method to recover them after the experiment.

The opportunity to explant and reuse Neuropixels probes is transformative. Given the cost ($1000 each, https://www.neuropixels.org/) and limited availability of the probes, many researchers will only be able to use them if it is possible to recycle them after experiments. The ability to recover these probes would enable researchers to repeat their experiments in different animals, boost the statistical power of their experimental findings, and thus enhance reproducibility of experimental data. We therefore sought to design a device for the Neuropixels probe that would allow experimenters to chronically implant it, run an experiment, and explant it for future experiments.

Several major innovations are required to design a removable holder for chronic implants of Neuropixels probes in unrestrained mice. First, the current design of the probe has several components that need to be securely mounted onto the small mouse skull. Further, these sensitive onboard electronics need to be protected while the mouse is in its home cage. Most importantly, the shank of the probe must be secured to ensure consistent recordings across weeks of recording. In previous work, this required permanently mounting the biosensor using adhesive, which was effective but made it nearly impossible to remove the probe afterwards (*Okun et al., 2016*). We also opted to use a 3D printed device in order to limit the use of acrylic in our design and ensure that it would be lighter than alternative designs.

To address these needs, we designed the Apparatus to Mount Individual Electrodes (AMIE), a device that fully encases and protects the sensitive onboard electronics of the Neuropixels probe, allowing long term, freely moving experiments. Moreover, the Neuropixels AMIE allows explantation and recycling. Our design and protocol is applicable to laboratories that wish to adapt the Neuropixels probe, or similar silicon probes, for recording in freely moving mice. With the drawings, materials and instructions, our device can be implemented not only by labs with years of expertise in electrophysiological recordings, but also by labs with different expertise that encounter a new need to study neural activity during behavior. Researchers that are using this technology in primates, rats, or in acute mice experiments may also find aspects of this approach useful.

With this design we have successfully recorded ~100 neurons simultaneously from unrestrained mice while observing freely moving behavior, and explanted the Neuropixels probe with a functioning recording shank. Further, because the AMIE is designed to allow implantation of a headbar (if desired), we recorded from the same mice in head-fixed experiments using systematic presentation of traditional visual stimuli. This feature of the AMIE allows experimenters to study neural activity in both psychophysical and ethological paradigms, affording the chance to build a bridge between the two.

## Results

### Design overview

The entire AMIE device weighs ~1.5 g (with cement:~2.0 g) and is assembled from three parts: the Neuropixels probe, the internal mount (IM), and external casing (EC) (*Figure 1A,B*; *Video 1*). The IM attaches directly to the Neuropixels PCB board with adhesive and is the core of the assembly

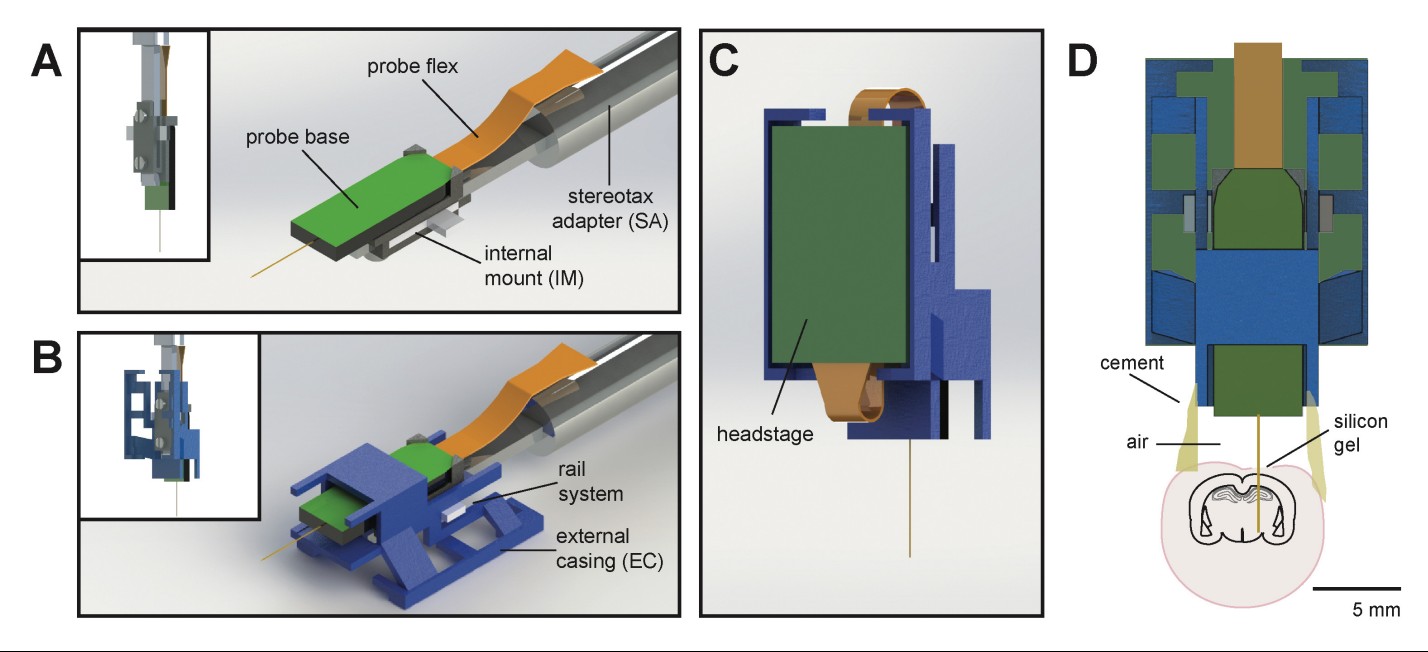

**Figure 1.** Schematic of Neuropixels AMIE. (**A**) Probe base mounted onto 3D printed internal casing and attached to machined metal stereotax adapter. Inset: Rear view, with screws that attach the internal mount (IM) to the stereotax adapter (SA). (**B**) Entire assembly in a. within 3D printed external casing. Inset: Rear view. (**C**) The headstage is positioned on the back of the encasing, with the flex wrapped in an 'S' shape. (**D**) Entire assembly in relation to size of mouse brain and skull. The EC is attached to the skull with cement. Silicon gel is used to as an artificial dura to protect the open craniotomy.
DOI: https://doi.org/10.7554/eLife.47188.002

(**Figure 1a**). On the backside of the IM is a slot for a stereotax adapter (SA) which allows for easy handling of the probe (**Figure 1A**). The IM attaches to the EC via a rail system (**Figure 1B**). During the implantation procedure, all adhesive binding the assembly to the rodent's skull exclusively contacts the EC, which acts as a protective shell (**Figure 1D**).

One difficulty in adapting the current Neuropixels design for freely moving experiments in mice is the ~3 cm long flex cable attached to a 1 g headstage (see **Jaeyoon et al., 2017**) for details). In early testing, we suspended the flex and headstage above the mouse's head during recording. However, we found that the flex very quickly twisted, potentially damaging it. In addition, the headstage added swinging weight above the mouse's head. With these observations in mind, we designed the encasing with a space for the headstage to be semi-permanently affixed. The probe flex wraps in an 'S' shape behind the implant, and attaches to the bottom (**Figure 1C**). In this way, the recording cable can be attached to the top of the implant, suspended above the mouse's head.

## Protocol overview

At least one day prior to implant, we attach the probe to the internal mount (**Figure 2A**). Silicone is

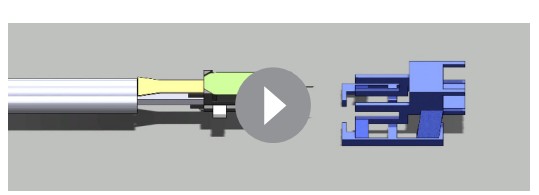

**Video 1.** 3D rendering of the AMIE device demonstrating the configuration of internal mount (IM), external casing (EC), and stereotax adapter.
DOI: https://doi.org/10.7554/eLife.47188.003

added to further secure the base of the recording shank (**Figure 2B**). Once this is dry, the internal mount is slid into the rails of the external casing and secured with cement (**Figure 2C,D**). This cement will be drilled away in order to explant the probe. When the entire AMIE assembly is dry, it is ready to be implanted (**Figure 2E**). The surgery to implant the probe and encasing typically takes ~3 hr (see Materials and methods for details). During this surgery, a headbar can also be implanted, which does not interfere with the encasing. The

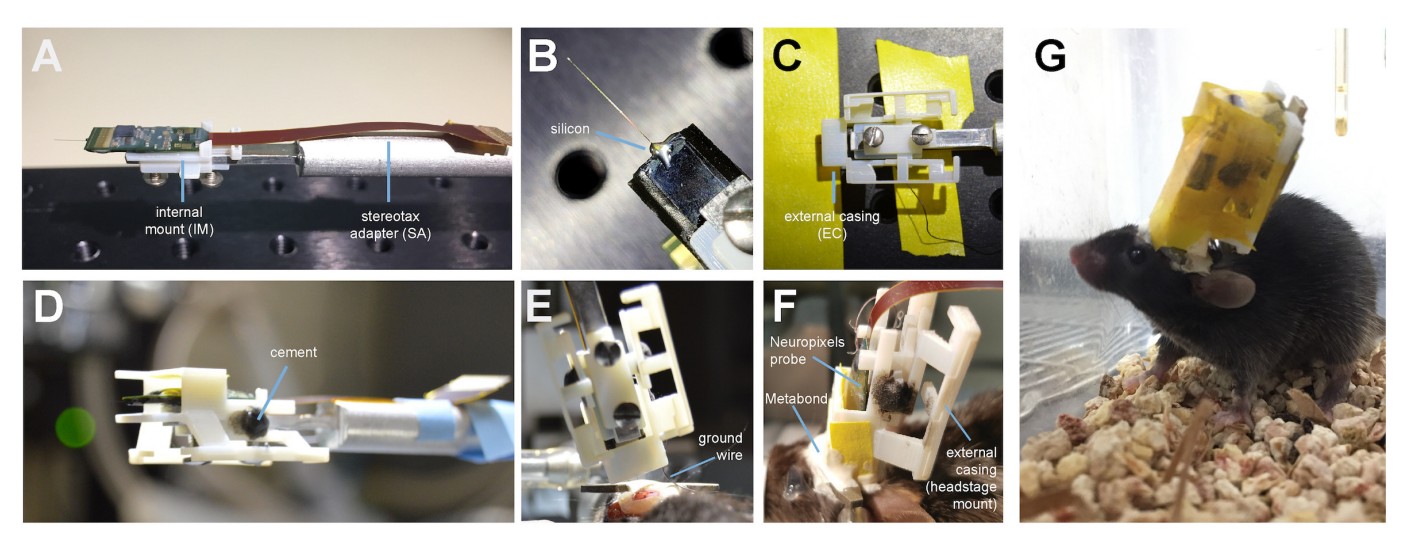

**Figure 2.** Mounting and implanting the Neuropixels probe. (**A**) The internal mount (IM) is attached to the stereotax adapter (SA) with two screws, and probe is attached to the internal mount using an epoxy. (**B**) Medical-grade silicon is added to the base of the shank to add extra support. (**C**) The external case (EC) is attached to a breadboard, and the IM+probe assembly is carefully guided into the internal compartment of the EC (top view). (**D**) After cementing the IM to the EC, the entire assembly is ready to be implanted. (**E**) During surgery, the the shank is lowered into the brain (here at a ~ 16˚ angle). The ground wire extends down the side of the implant and is attached to the ground screw. (**F**) The entire encasing is attached to the headbar and skull using Metabond. Tape is added where necessary to add protection between the encasing and the skull. The stereotax adapter (not shown) is removed after this support structure is dry and secure. (**G**) Image of a mouse with the implant ~48 hr after surgery. The entire assembly is wrapped in Kapton tape to protect the onboard electronics.

DOI: https://doi.org/10.7554/eLife.47188.004

external casing is the only part of the assembly that is attached to the skull (*Figure 2F*). In a typical experiment, we implant the probe and encasing without the headstage attached. We wait ~3–4 days for the mouse to recover, and then add the headstage. The headstage can be removed after each experiment, if desired. After ~1 day of habituation to the additional weight of the headstage (~1 g), we begin recording during behavior.

## Mice are mobile with the implant

Neuropixels probes were not designed for chronic implants in freely moving mice, and the entire probe assembly is quite bulky in comparison to a mouse's head (*Figure 1D*; *Jaeyoon et al., 2017*). However, we have designed a very slim encasing for the probe, and mice can adjust to the weight and size of the implant (*Video 2*).

By approximately 48 hr post-surgery, mice were mobile with the Neuropixels AMIE (*Figure 2G*). To evaluate the suitability of the AMIE for use during behavior, we assessed the impact of the device on both spontaneous and stimulus-driven movements. For spontaneous behavior, we analyzed video data taken while mice explored an open arena (*Figure 3A,B*). Even while tethered, implanted mice were typically agile and active (*Video 2*). To quantify behavior and compare for implanted vs. naive mice, we calculated three metrics from video data: the percentage of time spent moving, the maximum velocity and the maximum acceleration. For all three metrics, considerable overlap was apparent in the distribution of values for

**Video 2.** Behavior of mouse implanted with Neuropixels AMIE. Mouse was free to move around a 16"x16' arena while implanted and tethered. Video is shown at 2x speed.

DOI: https://doi.org/10.7554/eLife.47188.005

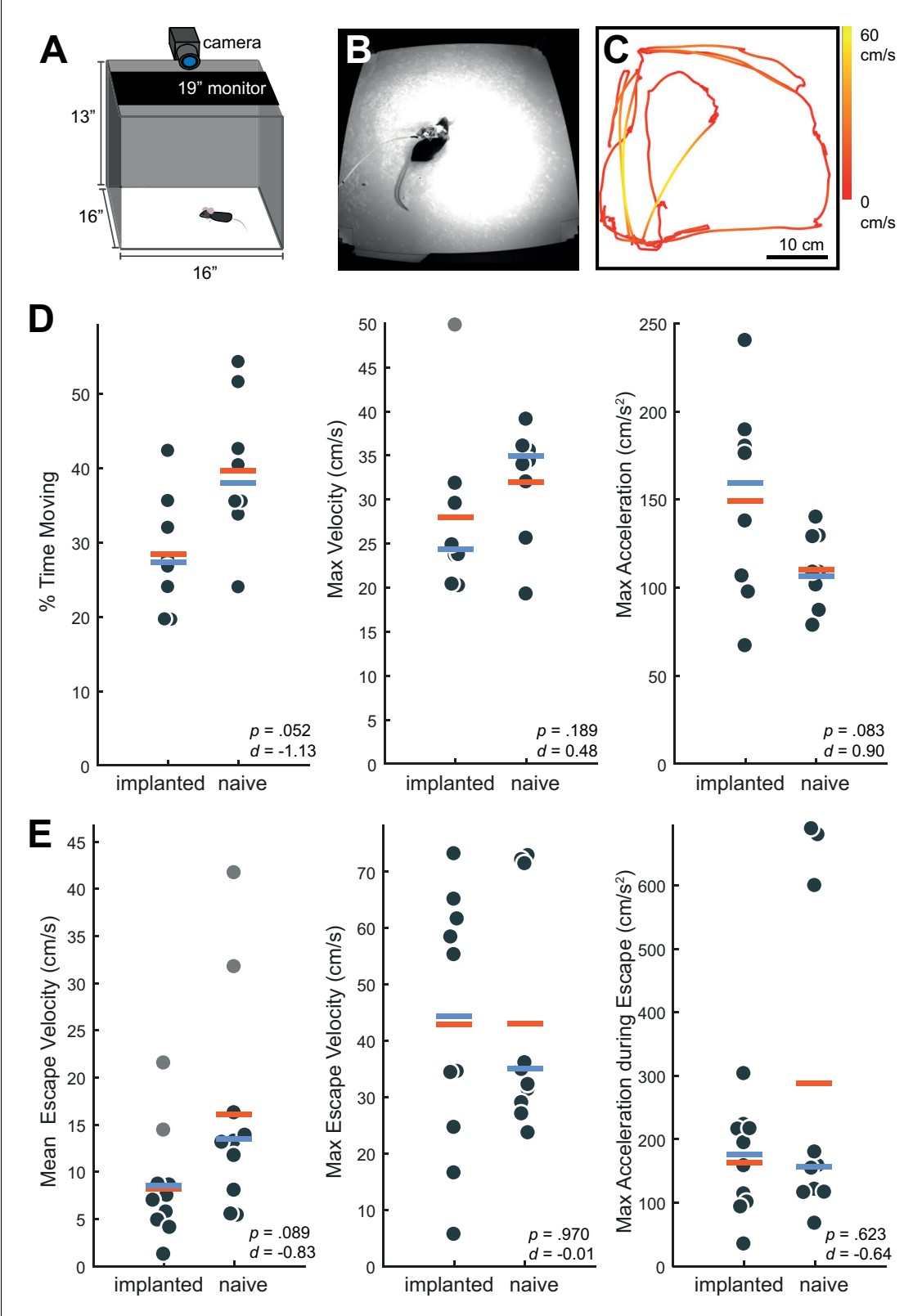

**Figure 3.** Behavior in implanted mice is comparable to naive mice. (A) Behavioral testing arena, with a camera to track the position of the mouse and a monitor on top to present visual stimuli. (B) Snapshot of mouse with implant in arena. (C) Sample tracking of 2 min of open field behavior in an implanted mouse. Color of the line indicates the velocity of the mouse. (D) Open field behavior of implanted vs. naive mice. Random 30–180 s exerpts of behavior (N = 8 videos per group, two videos from each mouse) in the open field were used to calculate a percent time moving (>5 cm/s), max

*Figure 3 continued on next page*

*Figure 3 continued*

velocity, and max acceleration. (**E**) Visual-looming evoked behavior of implanted vs. naive mice (N = 10 trials, two videos per mouse). A dark dot of linearly increasing diameter (40 cm/s) was presented over the mouse's head to evoke an escape response. The mean velocity, max velocity, and max acceleration during these responses is presented here. In all panels, orange line indicates the group mean, blue line indicates the median. p-Values (as computed by a two-sided Wilcoxon Rank Sum test) as well as effect sizes (computed by a Cohen's *d*) are reported on each panel. Outliers (defined as 1.5*IQR) are marked as light gray points.

DOI: https://doi.org/10.7554/eLife.47188.006

The following source data is available for figure 3:

**Source data 1.** Behavioral data for implanted and naive mice in an open field and in response to a looming stimulus.

DOI: https://doi.org/10.7554/eLife.47188.007

implanted and naive mice (*Figure 3D*). Although implanted mice moved slightly less and were slightly slower, the differences failed to reach significance for any metric. In fact, the mouse with the highest max acceleration was implanted (*Figure 3D*, middle panel).

To examine stimulus driven behavior, we measured responses to overhead visual looming stimuli (*Figure 3E*), which are known to elicit strong escape responses in mice (*De Franceschi et al., 2016*; *Evans et al., 2018*; *Yilmaz and Meister, 2013*). The distribution of values for the metrics tested (mean/max velocity and max acceleration) again overlapped considerably for naïve vs. implanted mice (*Figure 3E*). Although we observed no significant changes, a few naive mice achieved max acceleration during their escapes at values unobserved in implanted mice (*Figure 3E*, right). A possible explanation is that naive mice were free from the weight of the device and thus were able to accelerate very quickly when motivated to do so by a threatening stimulus. Taken together, these behavioral observations argue that although the presence of the AMIE may have idiosyncratically slowed mice slightly, they remained active in an open arena and showed species-typical responses to threatening stimuli.

## Neuropixels AMIE allows for 60–100 simultaneously recorded neurons across weeks of freely moving behavior

We recorded spiking activity across multiple brain areas during freely moving behavior over the course of 1–2 weeks. *Figure 4* illustrates an experiment with the probe implanted in medial visual cortex, subiculum, and midbrain. We isolated ~60–100 units for each session in this experiment (*Figure 4D,E*), during which the mouse moved freely around the arena and was exposed to looming stimuli. The number of single units we were able to isolate ranged across mice and experiments from ~20 to 145, but these numbers were fairly consistent within each mouse across recording sessions (*Figure 4C*). This variability is likely dependent on the probe that was used (Phase 3A Option four probes used in mouse #3 and #4 had 270 rather than 374 recordable channels; see Materials and methods and *Jun et al., 2017*), recording noise, and brain region. The absolute number of isolated units depends on the quality of the sorting and the experimenter's manual curation of Kilosort output, which does present challenging edge cases and can be difficult to assess with drift in the experiment. Overall, these numbers are less than has been previously reported with acute experiments in mice (*Jun et al., 2017*), possibly because of the chronic recording environment or inability to completely reduce noise. The longest we left a probe in was 41 days, without any noticeable decay in the signal.

To test how automatic unit sorting and classification would compare with our approach, we also sorted one of these freely moving sessions with Kilosort2, which automatically classifies units as 'good.' Indeed, for mouse #7, Kilosort2 identified 77 well-isolated units, compared to 69 with Kilosort1 and manual post-Kilosort designations in phy, confirming that our manual criteria were effective.

We elected to be conservative about any claims that the same neurons were recorded across days of the experiment, because demonstrating a stable recording of the same neurons from day-to-day is difficult and often regarded with skepticism. However, we did indeed observe waveforms that were consistent in both shape and depth across recordings, and it is entirely possible that these originate from the same neurons (*Figure 4F*).

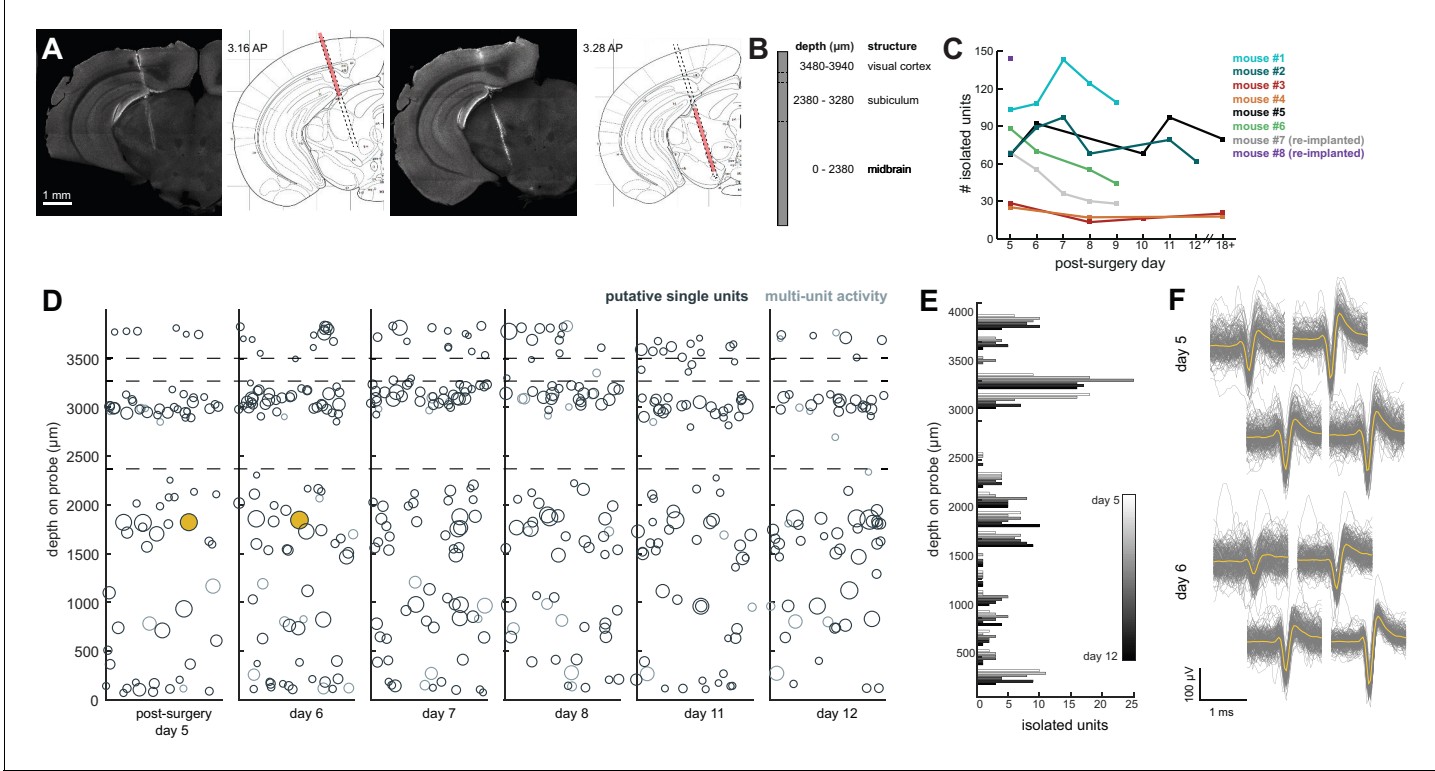

**Figure 4.** Chronic Neuropixel probe implants in cortex and subcortical regions can record ~20–145 units across multiple days. (A) Probe location, marked with DiI. Sections from Paxinos and Franklin atlas provided for reference. Mouse #200 was implanted with a probe in visual cortex, hippocampus (subiculum), and the midbrain. (B) Schematic of probe depth in (A). (C) Number of isolated units across recording days for eight different mice. Mouse #3 and #4 were implanted with a probe with fewer recording sites (270 vs. 374). Mouse #2 is featured in the other panels of this figure. Mouse #7 had a probe that was previously implanted in Mouse #5; see *Figure 6*. (D) Scatter plot of units across days for Mouse #2. Size of circles denotes number of waveforms assigned to that unit. X axis is random for visualization. € Histogram of isolated units across days and brain depth for Mouse #2. (F) Waveforms (n = 200, mean waveform in yellow) recorded from the same four contacts on the probe on day 5 (top) and day 6 (bottom). Units are the same as the yellow filled in circles in (D).

DOI: https://doi.org/10.7554/eLife.47188.008

The following source data is available for figure 4:

**Source data 1.** Number of isolated units for each probe across post-surgery days.

DOI: https://doi.org/10.7554/eLife.47188.009

## Researchers can also conduct headfixed recordings to further characterize neurons

A major limitation of many chronic implant designs is that they do not enable researchers to also implant a headbar to restrain the animal. The ability to head-fix animals critical for two reasons. First, it allows the experimenter to easily restrain the mouse during experiments, for example to attach/replace the headstage or fix twisting in the tether. Moreover, it affords the opportunity to measure neural activity in response to traditional psychophysical stimuli after the freely moving recording (*Figure 5*). This makes it possible to connect the neural responses obtained during an unrestrained, ethological task with those obtained during more traditional sensory electrophysiology context (simple stimuli defined by parameters that are systematically varied). This opportunity could prove invaluable in bridging observations from these two very different contexts which are normally studied in separate laboratories.

For example, after six days of recording freely moving behavior, we presented a battery of visual stimuli while the mouse was head-fixed to determine whether cells were visually responsive (*Figure 5A*). We were able to isolate 60 units (63 with Kilosort2) in the restrained condition, just as in the freely moving condition (*Figure 5B*). The distribution of units was similar to previous experiments where the mouse was not restrained.

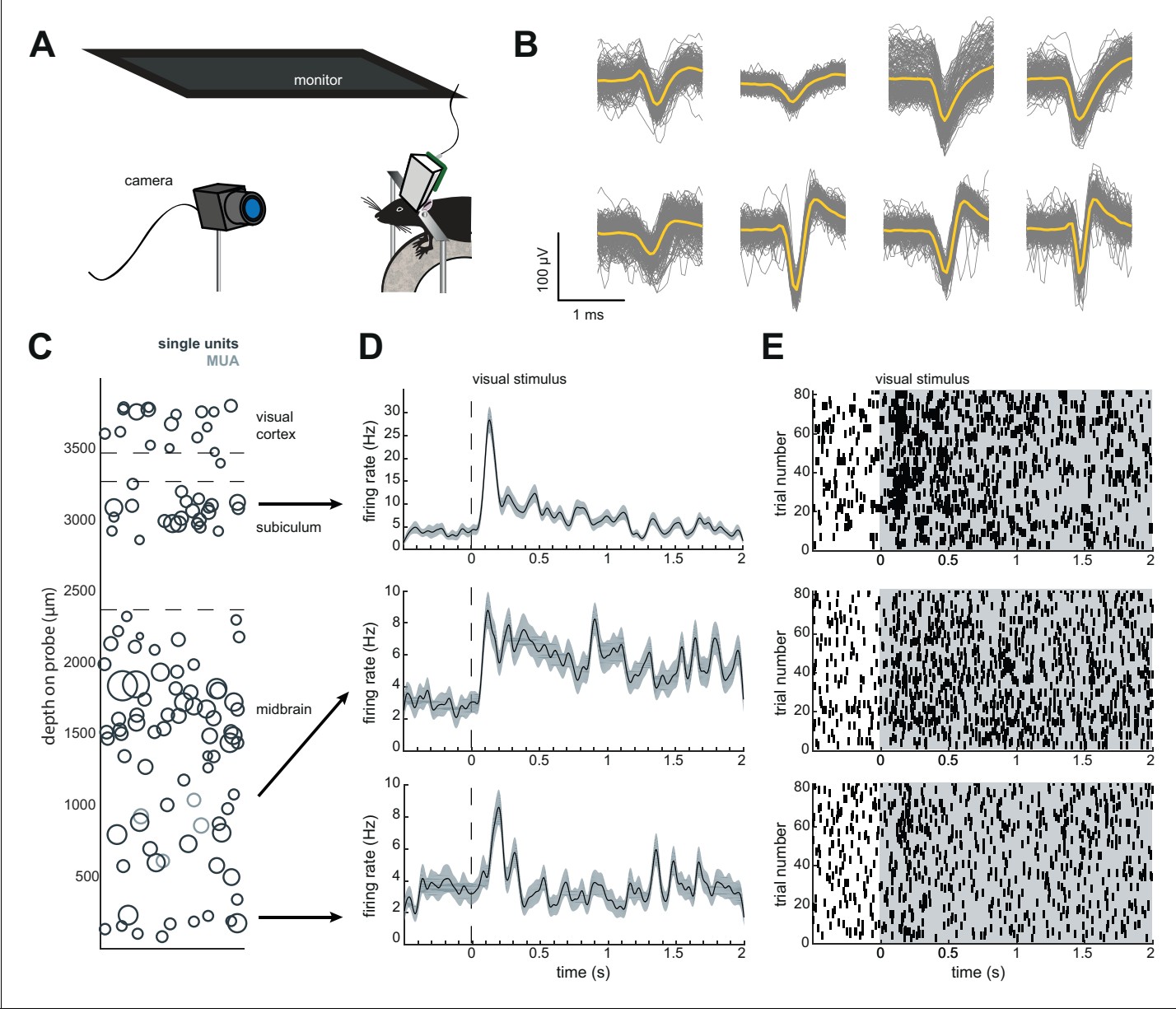

**Figure 5.** Implant design enables researchers to further characterize brain regions recorded during freely moving behavior. (**A**) Schematic of headfixed setup. The mouse was implanted with a headbar (see Materials and methods) enabling it to be restrained above a wheel. Visual stimuli was presented on a monitor above the mouse's head (similar to the unrestrained condition). The mouse's pupil can be tracked with a high resolution IR camera, and movement can be tracked using a rotary encoder on a 3D printed wheel. (**B**) Eight sample waveforms (n = 200, mean in yellow) from a restrained recording, same mouse as *Figure 4D–F* (Mouse #2). (**C**) Distribution of sorted units across the probe, same mouse as in *Figure 4D–F* (Mouse #2). (**D**) Peristimulus time histograms for three example neurons from different locations on the probe. The stimuli were a pseudorandomized set of 2 s full contrast sinusoidal drifting gratings in eight different directions. Shaded region is standard error of the mean. Stimuli began at the dotted line. (**E**) Raster plots for the neurons in (**D**). Shaded area indicates the duration of the stimulus.

DOI: https://doi.org/10.7554/eLife.47188.010

## Implant allows researchers to recover the probe after the experiment

Beyond providing a stable implant over many days, we also sought to design an implant that would allow for recycling of the Neuropixels probes. As demonstrated in *Figure 1*, the internal mount is separate from the external casing that is cemented to the mouse. After the completion of the experiment, researchers can drill away the cement and slowly remove the probe (see

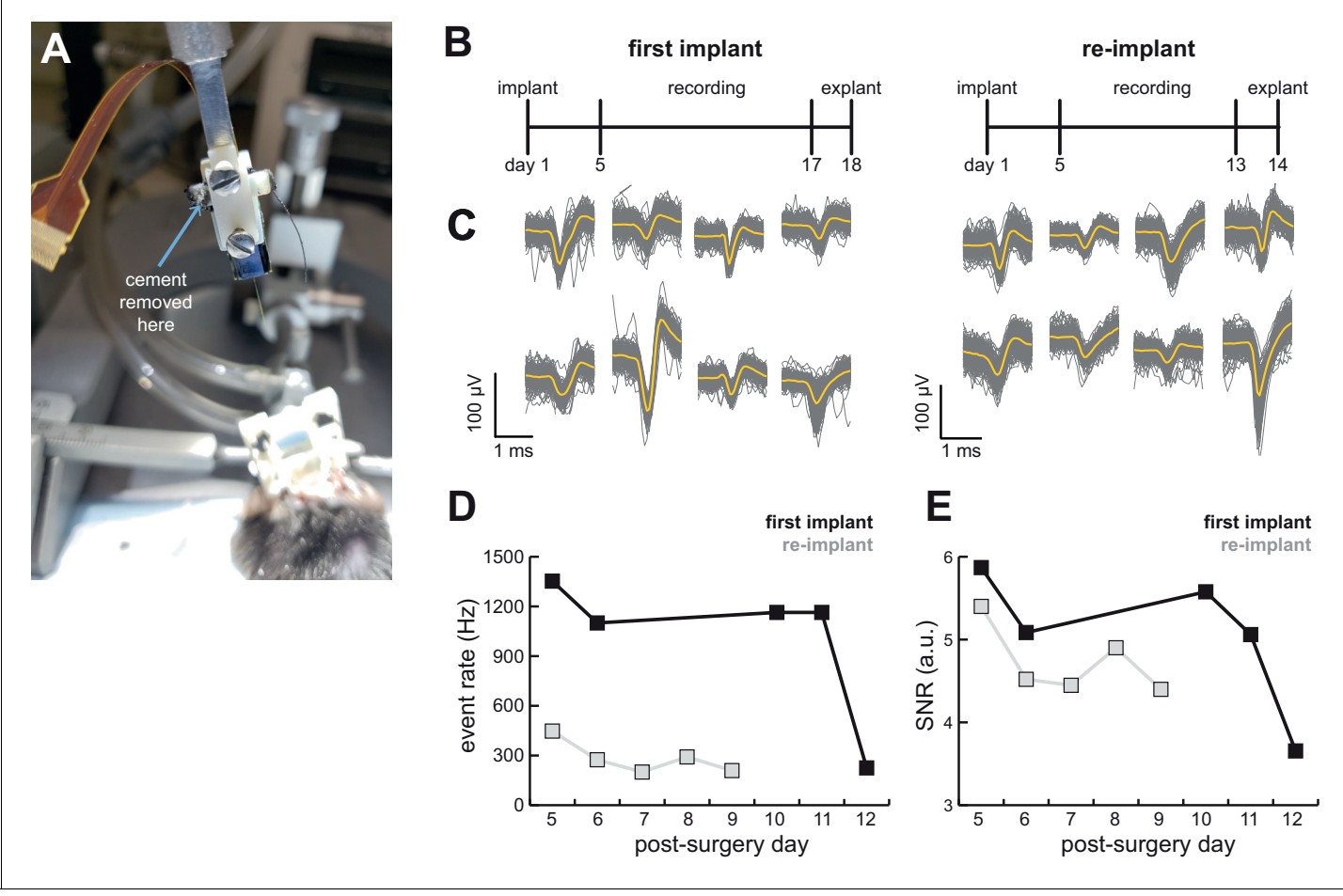

**Figure 6.** AMIE design allows for probe explantation and subsequent re-implantation with the same Neuropixels probe. (A) Example successful probe explanation. Cement is drilled away from the wings of the internal casing in order to remove the internal mount from the external casing. (B) Outline of experiment timing. The same probe was used in the first implant and re-implant. (C) Sample mean waveforms (n = 200, mean in yellow) from each mouse. (D) Detected event rate of the first implant versus the re-implanted probe across days of recording. (E) Median signal-to-noise ratio (SNR) for first implant and re-implanted probe across days (see Materials and methods.).
DOI: https://doi.org/10.7554/eLife.47188.011
The following source data is available for figure 6:

**Source data 1.** Computed event rate and SNR ratio for the initial implant and re-implant of the same probe across post-surgery days.
DOI: https://doi.org/10.7554/eLife.47188.012

Materials and methods and *Figure 6A*). This same probe, still attached to the internal mount, can then be re-secured within an external casing and implanted in another mouse.

We were able to record from a mouse for over 2 weeks, explant the probe, and re-implant for a second experiment (*Figure 6B*). We were easily able to isolate clear units in both (*Figure 4C and 6C*). Although we initially were able to isolate comparable numbers of units to the probe's first implant, the number of isolated units fell over time (compare mouse #5 and #7 in *Figure 4C*). Still, the re-implanted probe yielded 43.6 ± 17.8 neurons which is ample for many studies, especially those conducted in labs for which Neuropixels probes are a scarce resource.

In another experiment, were able to explant a probe from a mouse that did not recover from surgery and reimplant it in a second mouse (see *Table 1*). Although this second mouse ultimately also had complications resulting from a poorly positioned ground wire, one successful session of recording yielded 145 units (*Figure 4C*; *Table 1*). Further experiments will determine the unit yields that can be typically expected following reimplantation.

**Table 1.** Overview of experiments, with the Neuropixels probe option used and the outcome of the experiment.

For each of these experiments, even the unsuccessful explants, neural data was obtained from the initial implant and recording sessions. For an explanation of the probe options, see Materials and methods. Starred mice are included in the paper; + sign indicates the experiment was sorted with Kilosort2; M = mean; SD = standard deviation.

| Mouse | Probe option | Recordable channels | M ± SD Isolated units | Silicone on shank | Outcome |
|---|---|---|---|---|---|
| NP6* (*Figure 4*; Mouse #3) | 4 | 276 | 19.8 ± 6.23 | No | Shank broke during explant |
| NP7* (*Figure 4*; Mouse #4) | 4 | 276 | 20.0 ± 4.36 | no | Shank broke during freely moving recording |
| NP8* (*Figure 4*; Mouse #2) | 1 | 384 | 77.2 ± 13.7 | no | Shank broke during explant |
| NP9* (*Figure 4*; Mouse #1) | 1 | 384 | 117.4 ± 16.3 | no | Shank broke during explant |
| NP11 | 1 | 384 | - | no | Shank broke during freely moving recording |
| NP12 | 3 | 384 | - | no | Mouse didn't recover from surgery, probe successfully explanted and re-implanted in NP13 |
| NP13 (*Figure 4*; Mouse #8) | 3 | 384 | 145$^{+}$ | yes | Ground wire issues after surgery; one session successfully recorded. Successful explant |
| NP14* (*Figure 4 and 6*; Mouse #5) | 3 | 384 | 80.6 ± 13.6 | yes | Successful explant, re-implanted in NP16 |
| NP15* (*Figure 4*; Mouse #6) | 3 | 384 | 64.3 ± 19.1 | yes | Successful explant |
| NP16* (*Figure 4 and 6*; Mouse #7) | 3 | 384 | 43.6 ± 17.8 | yes | Successful explant |

DOI: https://doi.org/10.7554/eLife.47188.013

To assess the stability of the probe and our ability to detect spikes, we computed the event rate (sum of temporally coincident spikes on a group of sites for which the maximum amplitude exceeds the threshold) and signal-to-noise (SNR; see Materials and methods) ratio for the first implant of the probe as well as its re-implantation in another animal. There was a drop in the event rate and a small drop in the SNR in the re-implanted probe (*Figure 6D,E*). However, even with the first implant there was a significant drop in both event rate and SNR on the 12th day of recording, suggesting that this may not be due to the re-implantation itself.

Successful explant of probes depended on several factors. First, applying silicone to the base of the shank to add extra support appears to be necessary (*Figure 1B*). With silicone added to the base of the shank, 4/4 explant attempts were successful, whereas 1/6 explants were successful without the silicone (*Table 1*). Second, careful alignment of the probe, internal mount, and external casing will help ensure that the shank is being removed at the appropriate angle. Third, we only had success with Phase 3A Option three probes, suggesting that it may be easier with these, possibly due to the fact that the recording shanks on these probes are longer (10 mm) than Option 1 (5 mm). Fortunately, the shank of Phase 3B probes (now on the market) is also 10 mm long.

## Discussion

Here, we present a significant advance in our ability to use and recycle high-density silicon probes such as Neuropixels. Our device, the AMIE, and accompanying methods, allow researchers to perform recordings in both restrained and unrestrained conditions, and critically, to explant and reuse probes after experiments. This approach will enable researchers to capitalize on important technological advances to understand the complexity of brain activity during ethological behaviors, and to bridge the gap between ethological and psychophysical behaviors (*Gomez-Marin et al., 2014*).

Although Neuropixels probes were not designed for unrestrained recording in mice, our AMIE customizes them so that they are ideally suited to this purpose. The AMIE has a slim enclosure for the probe as well as the headstage (*Figures 1* and *2*), that mice can easily handle (*Figure 3*). It is worth noting that the Neuropixels design featured here is 3A, but the 3B (Neuropixels 1.0) version is the one currently commercially available. AMIE designs for both probe generations are available in the resources for this paper (see Materials and methods). Our design can also be readily adapted to other types of silicon probes (e.g. Neuronexus).

Unlike other electrophysiology systems, the current Neuropixels recording tether is not easily commutated due to heavy data demands. While this has not been a problem for recording from chronically-implanted rats in large arenas (*Jun et al., 2017*), it can be challenging for recordings from mice in smaller arenas, requiring constant monitoring of the mouse's position and occasional intervention from the experimenter to untangle the cord. In our experience, this is manageable, requiring the experimenter to stop an hour-long session once or twice to unplug the cord and untangle. Importantly, here we report similar behavior both during open field exploration and looming-evoked escape responses in implanted and naive mice (*Figure 3*).

The Neuropixels AMIE can be used to record in both restrained and unrestrained conditions, with similar yields in numbers of isolated units (*Figures 4* and *5*), although a direct comparison of yields across publications is challenging due to potential differences in spike sorting criteria across labs. The ability to restrain the mouse for passive stimulation enables researchers to obtain additional information about their recordings that may ultimately aid in uncovering the function of cells and brain regions. Remarkably, during our headfixed experiments we found that even cells deep in the midbrain showed clear visual responses to drifting gratings (*Figure 5D,E*). This demonstrates the power of Neuropixels to uncover signals relevant to decision-making and other behaviors in uncharted brain territories.

## Materials and methods

**Key resources table**

| Reagent type/resource | Designation | Source or reference | Identifiers | Additional information |
|---|---|---|---|---|
| Chemical compound/drug | Medical-grade clear silicon adhesive | Mastersil | 912MED | |
| Chemical compound/drug | Loctite Instant Adhesive 495 | ULINE | S-7595 | |
| Chemical compound/drug | Medigel CPF | Clear H20 | 74-05-5022 | |
| Chemical compound/drug | Isoflurane | Allivet | 50562 | |
| Chemical compound/drug | C and B Metabond 'B' Quick Base | Parkell | S398 | |
| Chemical compound/drug | C and B Metabond 'C' Quick Base | Parkell | S371 | |
| Chemical compound/drug | C and B Metabond Radiopaque L-Power | Parkell | S396 | |
| Chemical compound/drug | Optibond Solo Plus | Kerr | 31514 | |

*Continued on next page*

*Continued*

| Reagent type/resource | Designation | Source or reference | Identifiers | Additional information |
|---|---|---|---|---|
| Chemical compound/drug | Vetbond | Santa Cruz Biotechnology | sc-361931 | |
| Chemical compound/drug | Charisma A1 Syringe | Net32 | 66000085 | |
| Chemical compound/drug | Eye Ointment | Rugby | 370435 | |
| Chemical compound/drug | Dental Cement | Stoelting | 5217307 | |
| Chemical compound/drug | DiI | ThermoFisher Scientific | D282 | |
| Chemical compound/drug | Silicone Gel Kit | Dow Coning | 3–4860. | |
| Chemical compound/drug | Bleach | Amazon | B01K8HT54G | Any brand bleach ok |
| Other | Neuropixel Probe | Neuropixel Stock Center (Neuropixels.org) | Neuropixel 1.0 Probe | |
| Other | 3D Printed Internal Mount | 'this paper' - Github repository | IM_Neuropixel1.stl | Internal mount design file (.stl) can be downloaded from the following github repository: https://github.com/churchlandlab/ChronicNeuropixels |
| Other | 3D Printed External Casing | 'this paper' - Github repository | EC_Neuropixel1.stl | same as above |
| Other | Sterotax Adapter | 'this paper' - Github repository | stereotax adapter v4.ipt | same as above |
| Other | 2-56A Screws | Amazon | B00F34U238 | |
| Other | Silver Wire | WPI | AGW1010 | |
| Other | 4' post holder with thumbscrew | Thorlabs | PH4 | |
| Other | Slim right angle bracket | Thorlabs | AB90B | |
| Other | Aluminum Breadboard | Thorlabs | MB624 | |
| Other | M6 Cap Screw | Thorlabs | SH6MS20 | |
| Other | M6 Nut | Thorlabs | HW-KIT2/M | |
| Other | Kapton Tape | ULINE | S-7595 | |
| Other | Kimwipes | Kimtech | 34120 | |
| Other | Oxygen Cylinders | Airgas | OX USP300 | |
| Other | Mouse Anesthesia System with Isoflurance Box | Parkland Scientific | V3000PK | |
| Other | Small rodent sterotax fitted with anesthesia mask | Narishige | SG-4N | |
| Other | Dental Drill | Osada | EXL-M40 | |
| Other | 0.9 mm burrs for micro drill | Fine Science Tools | 19007–09 | |
| Other | T/Pump Warm Water Recirculator | Kent Scientific | TP-700 | |

| Reagent type/resource | Designation | Source or reference | Identifiers | Additional information |
|---|---|---|---|---|
| Other | Warming Pad for warm water recirculator | Kent Scientific | TPZ | |
| Other | Cotton Applicators | Fisher Scientific | 19-062-616 | |
| Other | Surgical Spears | Braintree Scientific Inc. | SP 40815 | |

## Printing and machining parts

To conduct this experiment, researchers will need Neuropixels probes. We recommend performing the entire process of preparing and implanting the probe using a dummy probe for practice. We printed and tested in VeroWhite material using a Stratasys Eden 260VS PolyJet 3D Printer with 16 μm resolution. The stereotax adaptor should be machined from aluminum or stainless steel. The parts featured here were designed for Neuropixels 3A probes, but we have since adapted these for Neuropixels 3B probes (Neuropixels 1.0). All designs can be found on the CSHL repository (http://repository.cshl.edu/36808/) as well as on Github (*Juavinett et al., 2019*; (copy archived at https://github.com/elifesciences-publications/ChronicNeuropixels).

The probe options for Neuropixels 3A differ based on their probe length (and corresponding site count), as well as whether they are active or passive electrodes. Probe options 1 and 3 are both passive, and contain 384 (5 mm long shank) and 960 sites (10 mm long shank), respectively. Probe options 2 and 4 are both active, and contain 384 (5 mm long shank) and 966 sites (10 mm long shank) respectively. All the options have the option to record from 374 channels, with the exception of Option 4, which only has 270 recording channels. Readers should refer to the Supplementary Information in *Jun et al. (2017)* for additional details. Neuropixels 3B probes have the Phase 3A Option three shank.

## Mounting the probe

First, the internal mount is secured to the stereotax adapter (SA) using two 2-56A screws (Amazon, B00F34U238). As depicted in *Figure 2A*, we then attached the Neuropixels probe to the internal mount (IM) usingLoctite Instant Adhesive 495 (ULINE S-17190). Using a needle, we applied a medical-grade clear silicone adhesive, Mastersil 912MED, to the base of the shank (*Figure 2B*). The IM and probe was slid into the rails of the external casing (EC), and secured with cement (*Figure 2D–F*).

## Surgical methods

All surgical and behavioral procedures conformed to the 316 guidelines established by the National Institutes of Health and were approved by the Institutional 317 Animal Care and Use Committee of Cold Spring Harbor Laboratory. We used male 3–4 month old C57/BL6 mice (Jackson Laboratories, 000664). Male mice were used because they are typically larger, and we expected that they would better handle the weight of the implant. Mice were given medicated (carprofen) food cups (MediGel CPF, Clear H20 74-05-5022) 1–2 days prior to surgery.

During surgery, the mouse was anesthetized with isoflurane. We cut away the skin and cleared any connective tissue. Tissue at the edges of the skull was glued down with Vetbond (Santa Cruz Biotechnology, cat. no. sc-361931). The skull was cleared and dried, using a skull scraper or blade to add additional texture. A boomerang shaped custom Titanium headbar was cemented to the skull, just posterior to the eyes, near Bregma. A burr hole was drilled for the ground screw, which was carefully screwed into the skull. We applied Optibond Solo Plus (Kerr, cat No. 31514) to the skull, and used UV light to cure it. We used Charisma (Net32, cat. No. 66000085) to create a base for the implant, and add additional support around the ground screw. Using a dental drill, a small craniotomy (1–2 mm) was made over visual cortex (2–2.5 ML,−3.4–3.5 AP relative to Bregma). The entire Neuropixels assembly (SA, IM, and EC) was placed in the stereotax. After carefully applying DiI (ThermoFisher Scientific, cat. no. D282, 0.5% in DMSO) to the probe, the shank was slowly lowered into the brain at a ~ 16 degree angle (*Figure 2E*). The ground wire is wrapped around the ground screw, and Metabond cement was carefully applied to attach the EC to the skull. The entire

assembly was wrapped in Kapton Tape (ULINE S-7595) and the mouse was allowed to recover for 3–4 days. Mice were housed individually after surgery.

Once the mouse recovered, we removed the tape and added the headstage to the back of the implant. The entire assembly was re-wrapped with tape. On the next day, we began behavioral testing.

### Behavioral data

To compare the behavior of implanted mice with naïve/unimplanted mice, we tracked mice using a Basler Pylon camera and Ethovision XT13 in a 16' x 16' open arena. For open field tests, naïve mice were allowed to explore a bare arena for 15 min. Implanted mice were tested in an arena with an inset nest; the data presented here are random excerpts of the mouse's activity while outside of the nest. We excerpted the same length time segments from the naive mice for comparison. Our behavioral data were not normally distributed, so a Wilson Rank Sum Test was used to test for differences between naïve and implanted mice. We computed effect sizes using Cohen's *d*.

### Visual stimulation

For visually-evoked responses during freely moving behavior (*Figure 4*), a linearly expanding dot (40 cm/s) was presented on a monitor directly over the mouse's head. This stimulus is known to elicit an escape response in mice (*De Franceschi et al., 2016*; *Yilmaz and Meister, 2013*). Unimplanted mice could escape into a small nest: a triangular prism with a 13 cm opening. Implanted mice could escape into a nest inset into the wall – this modification was necessary to enable mice to enter the enclosure with the implant. We found that being able to easily enter the nest increased the probability of flight (vs. freezing) responses. For visually evoked responses during head restraint (*Figure 5*), a set of full contrast, full field drifting gratings in eight different directions (10 repeats) were presented above the mouse's head while the mouse was free to move on a wheel.

### Electrophysiology data

Electrophysiology data was collected with SpikeGLX (Bill Karsh, https://github.com/billkarsh/SpikeGLX). The data were first median subtracted across channels and time (see *Jun et al., 2017*). Unless otherwise noted, experiments were first sorted with Kilosort spike sorting software (*Pachitariu et al., 2016*) and manually curated using phy (https://github.com/kwikteam/phy). Numbers of recorded neurons here may be more conservative than previously published reports because we were careful to exclude any units that exhibited drift or had evidence of being more than one neuron. Specific experiments (as noted in the text) were sorted with Kilosort2 for comparison (https://github.com/MouseLand/Kilosort2). Additional analyses and plotting with data were done with MATLAB code modified from N. Steinmetz (https://github.com/cortex-lab/spikes). To assess the quality of our recordings, we computed two metrics. First, we calculated the rate of spikes above the noise floor ('event rate'). Events are temporally (<1 ms) and spatially (~50 μm radius) consistent events with amplitudes (on any site) that exceed six times the median absolute deviation (MAD, *Jun et al., 2017*). In addition, we computed the signal-to-noise ratio SNR for each event. As previously described, the event SNR is the ratio of peak amplitude of the site with largest amplitude (negative peak) in the event to $0.6745 \times$ MAD (*Jun et al., 2017*).

### Probe explantation

To explant the probe, we first anesthetized the mouse with isoflurane and loosely positioned the mouse into the earbars. The SA was placed in the stereotax and aligned with its slot in the IM. We carefully lowered the SA into the IM, and put the two screws back into place. It was important that the SA was properly aligned with the IM so that no unnecessary tension was placed on the implant. We carefully drilled away the cement at the boundary of the IM and EC, unraveled or cut the ground wire, and slowly raised the SA+IM+probe assembly. The mouse was perfused and the brain was fixed in 4% PFA for sectioning. We were able to find DiI signals in the brain even 1 month after implantation (we did not test later time points).

A detailed surgical protocol for mounting, implanting, and explanting the probe is located on Github (*Juavinett et al., 2019*; copy archived at https://github.com/elifesciences-publications/ChronicNeuropixels).

## Acknowledgements

This work represents the collective input and knowledge of a burgeoning Neuropixels community (http://www.neuropixels.org).

We would like to acknowledge Tim Harris for his leadership on the development of the Neuropixels probes and his constant encouragement of this project.

The UCL Neuropixels course, taught by Nick Steinmetz, Matteo Carandini, Andrew Peters, Adam Kampff, was imperative in getting this project off the ground (http://www.ucl.ac.uk/neuropixels/courses). In particular, we would like to thank Nick Steinmetz for his critically important feedback, code, and upkeep of the Neuropixels Wiki page (https://github.com/cortex-lab/neuropixels/wiki).

We would also like to acknowledge Claudia Boehm and Albert Lee (Janelia Research Campus) for allowing us to observe their rat Neuropixels implant. Their protocol served as an important starting point for the protocol we developed in mice.

We have also benefitted from troubleshooting help from many individuals, including Wade Sun, James Jun, Marius Bauza, and Bill Karsh (SpikeGLX).

We are also grateful to the CSHL Undergraduate Research Program, which yearly provides a diverse group of students with funding and resources to complete invaluable research experiences at CSHL. This program funded GB for his initial summer in our lab.

We welcome feedback from the community regarding the diversity of methods used to implant and record with these probes.

## Additional information

### Funding

| Funder | Grant reference number | Author |
| --- | --- | --- |
| Simons Foundation | Simons Collaboration on the Global Brain | Anne K Churchland |
| Pew Charitable Trusts | Pew Scholars | Anne K Churchland |
| Eleanor Schwartz Fund | Scholar Award | Anne K Churchland |
| Cold Spring Harbor Laboratory | Marie Robertson | Anne K Churchland |
| National Science Foundation | 1559816 | George Bekheet |

The funders had no role in study design, data collection and interpretation, or the decision to submit the work for publication.

### Author contributions

Ashley L Juavinett, Conceptualization, Data curation, Software, Formal analysis, Supervision, Investigation, Visualization, Methodology, Writing—original draft, Writing—review and editing; George Bekheet, Data curation, Software, Formal analysis, Investigation, Visualization, Methodology, Writing—original draft, Writing—review and editing; Anne K Churchland, Conceptualization, Resources, Supervision, Funding acquisition, Methodology, Writing—original draft, Project administration, Writing—review and editing

### Author ORCIDs

Ashley L Juavinett (ID) https://orcid.org/0000-0002-4254-3009
Anne K Churchland (ID) https://orcid.org/0000-0002-3205-3794

### Ethics

Animal experimentation: All surgical and behavioral procedures conformed to the guidelines established by the National Institutes of Health and were approved by the Institutional Animal Care and Use Committee of Cold Spring Harbor Laboratory (protocol # 16-13-10-7). All surgery was performed under isoflurane anesthesia and every effort was made to minimize suffering.

**Decision letter and Author response**
Decision letter https://doi.org/10.7554/eLife.47188.019
Author response https://doi.org/10.7554/eLife.47188.020

# Additional files

## Supplementary files
• Transparent reporting form
DOI: https://doi.org/10.7554/eLife.47188.014

## Data availability

We have made all the materials related to this device available to the community via GitHub (https://github.com/churchlandlab/ChronicNeuropixels; copy archived at https://github.com/elifesciences-publications/ChronicNeuropixels). The technical drawings, the methodological instructions, the photographs and supporting code will, together, allow any researcher to rapidly adopt this new technology and begin to benefit from Neuropixels probes. Data from the electrophysiological recordings available is available here: http://churchlandlab.labsites.cshl.edu/code/.

The following dataset was generated:

| Author(s) | Year | Dataset title | Dataset URL | Database and Identifier |
|-----------|------|---------------|-------------|-------------------------|
| Ashley L Juavinett, George Bekheet, Anne K Churchland | 2019 | Chronically-implanted Neuropixels probes enable high yield recordings in freely moving mice: dataset | http://repository.cshl.edu/38304/ | CSHL Institutional Repository, 38304 |

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
