## [Decision Letter]

Thank you for submitting your article "Chronically-implanted Neuropixels probes enable high yield recordings in freely moving mice" for consideration by *eLife*. Your article has been reviewed by three peer reviewers, and the evaluation has been overseen by Laura Colgin as the Senior and Reviewing Editor. The following individual involved in review of your submission has agreed to reveal their identity: Nick Steinmetz (Reviewer #1).

The reviewers have discussed the reviews with one another, and the Senior Editor has consolidated their comments and discussion into this decision letter to help you prepare a revised submission.

Summary:

The authors have presented a method and specifications for achieving chronic Neuropixels recordings in freely-moving mice and, importantly, for recovering the probes afterwards. Due to the financial cost and limited availability of Neuropixels probes, chronic implantation of these probes is currently challenging (i.e., researchers may not want to "waste" a probe by chronically implanting it). This paper presents a potential solution to this problem. Many labs have ordered Neuropixels probes since they were made available, so reviewers agreed that these methods provide a useful resource for many in the community. The proposed system consists of two main components: 1) an internal mount permanently fused to the probe base and 2) a removable casing (Apparatus to Mount Individual Electronics: AMIE) that adapts the probe and internal mount to a chronic implant. In addition to carrying the probe, the AMIE protects and holds the flex-cable and head stage. Reviewers found the paper to be clear and appreciated the authors' honesty about potential pitfalls (e.g., broken probes, early attempts with low yields). Reviewers appreciated that the authors made their surgical protocol and 3D printer files publicly available. However, a number of major concerns were raised that need to be addressed before the paper can be deemed suitable for publication.

Essential revisions:

1) Reviewers felt that the manuscript misses key data showing that the Neuropixel can be re-used without a significant drop in signal quality. Specifically, in Table 1 average numbers of isolated units are missing for NP13, 15 and 16 (i.e. the cases in which Neuropixels were implanted and then successfully explanted and re-implanted in a second mouse). These numbers are key (particularly NP16), as the authors claim that the "quality of the recording did not noticeably change in the second mouse", but do not provide any quantitative data to support their claim. Instead, they show 5 example units from NP14 and 16, which only demonstrates they could find at least 5 units in both recordings (Figure 6C). Additional quantification would also be bolster the manuscript's reusability claims. It would be useful to compare noise levels on the first implantation to that on the second for a given probe. This can be computed straightforwardly e.g. with this function (https://github.com/cortex-lab/spikes/blob/master/analysis/computeRawRMS.m). The authors should also run kilosort2 on all recordings (https://github.com/MouseLand/Kilosort2), which automatically assigns 'good' and 'mua' labels. This is not necessarily perfectly accurate, but at least it would provide a rapid and unbiased analysis that would be comparable to yields reported in other recordings (e.g. comparable to this dataset https://janelia.figshare.com/articles/Eight-probe_Neuropixels_recordings_during_spontaneous_behaviors/7739750). Alternatively, the authors could repeat the manual process that they did for some recordings already for at least one recording from each of the other mice in Table 1. The same argument applies to the comparison between restrained and unrestrained, which needs similar automatic (i.e., KS2) or manual quantification or both. The reusability of the probes is critical for the paper's significance, because (as the authors note) techniques without explanation already exist to chronically implant Neuropixels in freely moving mice and re-use (Introduction, fifth paragraph, Okun, 2016).

2) A further concern is how the authors demonstrate that behavior in implanted vs. naïve mice is comparable. Quantitatively, they do this by reporting non-significant p-values on rank-sum tests of differences between behavioral measures in implanted vs. naïve mice. However, using non-significant p-values to argue for the null-hypothesis is not statistically valid; failing to reject the null hypothesis is not evidence for the null hypothesis. It's just as likely that the authors didn't use enough mice to have sufficient statistical power to show the effect. Indeed, visual inspection of Figure 3 makes this appear highly likely – clear changes in several activities can be seen in Figure 3 (most notably for % time moving, max acceleration and mean escape velocity). For instance, time moving declines from about 40% to 30%, and it's hard to believe that with additional samples this would remain non-significant, especially given that p = .0502. Relatedly, the authors claim that the correct critical value should be p = .0167 due to multiple comparisons. The meaning of a multiple-comparisons correction is difficult to interpret when the researcher's goal is to show that a difference isn't significant. Then, one could perform rank-sum tests on an arbitrarily large number of behavioral measures to reduce the critical value, and find no differences whatsoever between populations. A simple and more interpretable approach would be to simply report the effect sizes instead of p-values. Though there is a small reduction in exploratory behavior/ escape velocity, this is relatively small and essentially unavoidable due to the weight of any probe/casing being placed on the mouse's head.

3) Currently shipping probes do not have the same geometry as the probes described in this manuscript. That is, the probes that are now on sale at neuropixels.org have a slightly different shape than the "Phase 3A" probes employed in this paper. The authors need to describe how their system could be adapted to other probe geometries (and future designs will almost certainly evolve in this regard). It would be extremely valuable if the authors could provide an updated design based on the changes to the probes. Even though it may only be subtly different, it will save others from having to replicate the work.

4) Reviewers had trouble visualizing the exact shape of the pieces and how they fit together. One suggestion was that the authors could make a short video that shows someone putting the pieces together by hand with a dummy probe. Some sort of additional visual aid such as this would greatly help readers to understand what screws onto what, how things slide, etc. Another suggestion was that it could perhaps somehow be done with the 3D renderings from Figure 1.

5) A concern was raised about whether the authors' system for recovering the probe might compromise the stability of the probe, since most of the body of the probe itself is not really directly attached to the skull, but only via a thin 3D printed piece. A potential way to test this that was suggested could be to plot a comparison of waveforms like in Figure 4F, but for the same neuron recorded during periods of high acceleration versus stationary periods. An even better test would be to compute a metric of waveform similarity between moving and stationary periods, which could be compared to a recording in which the probe was attached in a fixed way. Perhaps such a recording would be available from Evans et al., who may have performed chronic recordings in freely moving animals with a probe fixed in place in the manner of Okun et al., 2016. Other reasonable ideas about how to address this concern would also be acceptable.

6) Some discussion of why this technology would be useful even if probes were less expensive and more accessible (or how/whether the approach would be modified under such conditions) would help extend the impact of this paper, assuming that limitations of the neuropixels purchasing scheme will be alleviated in the future.

---

## [Author Response]

Essential revisions:1) Reviewers felt that the manuscript misses key data showing that the Neuropixel can be re-used without a significant drop in signal quality. Specifically, in Table 1 average numbers of isolated units are missing for NP13, 15 and 16 (i.e. the cases in which Neuropixels were implanted and then successfully explanted and re-implanted in a second mouse). These numbers are key (particularly NP16), as the authors claim that the "quality of the recording did not noticeably change in the second mouse", but do not provide any quantitative data to support their claim. Instead, they show 5 example units from NP14 and 16, which only demonstrates they could find at least 5 units in both recordings (Figure 6C). Additional quantification would also be bolster the manuscript's reusability claims.

We appreciate this concern from the reviewers. We have added the number of isolated units for our re-implanted probe (NP16/Mouse #7) as well as an additional mouse (NP15/Mouse# 6) to Figure 4C, and have added the average number of isolated units to Table 1. Although the probes in NP13 and NP15 were successfully explanted, they were not implanted in a second mouse.

It would be useful to compare noise levels on the first implantation to that on the second for a given probe. This can be computed straightforwardly e.g. with this function (https://github.com/cortex-lab/spikes/blob/master/analysis/computeRawRMS.m).

We wholeheartedly agree. We have added new panels to Figure 6 that address the concern about noise levels on the probe. These panels (6D and 6E) show the detected event rate and SNR for the re-implanted probe (using the recommended code and following the example in Jun et al., 2017). Although there is a drop in detected event rate and a slight drop in SNR in the re-implanted probe, we note that it is still possible to isolate many units (43.6 ± 17.8). Two points should be considered when evaluating this yield. These are described below.

First, variability across yields in individual animals even for a new probe (Figure 4C) leaves open the possibility that yields on re-implanted probes might sometimes be higher than what we observed. To test this, we sorted an additional, previously unsorted dataset from a new mouse (NP13) which had a probe that was re-implanted after being in NP12. We had not originally included NP13 in this dataset because of issues with the grounding wire and screw after surgery, enabling us to only record one session. We are pleased to report that sorting of this session yielded 145 well-isolated, ‘good’ units. This number is now included on Figure 4C, highlighting that high yields are possible even for re-implanted probes.

Second, yields of ~40 neurons, while not ideal, are likely acceptable to many researchers because Neuropixels probes are still available in very limited numbers. The recent release of 3B probes from IMEC was a first step towards increased availability, but even many well-established systems neuroscience labs were only granted 10/year (indeed, the last author of this paper was provided with that number). This is a small number even for a single project, especially one that involves a trainee who will have to learn to use the probes and may even break a few. For such labs, which are probably the vast majority in Neuroscience, a yield of 43 simultaneously recorded neurons for a re-implanted probe has tremendous value.

The corresponding results text can be found in the subsection “Printing and machining parts”. We thank the reviewer for pushing us to include additional data on yields and feel that this has greatly strengthened the paper.

The authors should also run kilosort2 on all recordings (https://github.com/MouseLand/Kilosort2), which automatically assigns 'good' and 'mua' labels. This is not necessarily perfectly accurate, but at least it would provide a rapid and unbiased analysis that would be comparable to yields reported in other recordings (e.g. comparable to this dataset https://janelia.figshare.com/articles/Eight-probe_Neuropixels_recordings_during_spontaneous_behaviors/7739750). Alternatively, the authors could repeat the manual process that they did for some recordings already for at least one recording from each of the other mice in Table 1. The same argument applies to the comparison between restrained and unrestrained, which needs similar automatic (i.e., KS2) or manual quantification or both. The reusability of the probes is critical for the paper's significance, because (as the authors note) techniques without explanation already exist to chronically implant Neuropixels in freely moving mice and re-use (Introduction, fifth paragraph, Okun, 2016).

We agree with the reviewer that the automatic assignment of ‘good’ and ‘mua’ labels could be useful in comparing of neuron yields. To evaluate whether Kilosort2 differed from our approach, we implemented it for two separate analysis. First, we implemented Kilosort2 on the first experiment with the re-implanted probe in NP16. Neuron yields with Kilosort2 were similar to our original sorting method, which was reassuring (77 for

Kilosort2 compared to 69 with our original sorting). The slightly higher yield for KS2 may reflect the conservativeness of our original approach. Second, we implemented Kilosort2 during a head-restrained experiment and again, our original yields were similar, if slightly more conservative (63 for Kilosort2 compared to 60 for our original approach). The similar neuron yields for our original approach and KS2 suggest that our criteria for spike sorting were effective: robust units with clear waveforms, reasonable interspike intervals, and minimal drift in depth. The similarity of our original approach to Kilosort2 made us disinclined to re-analyze all the datasets used for the comparisons in the paper. Nevertheless, we include the numbers above in the revised manuscript (subsection “Researchers can also conduct headfixed recordings to further characterize neurons”) as readers may find these observations useful in evaluating spike sorting methods.

2) A further concern is how the authors demonstrate that behavior in implanted vs. naïve mice is comparable. Quantitatively, they do this by reporting non-significant p-values on rank-sum tests of differences between behavioral measures in implanted vs. naïve mice. However, using non-significant p-values to argue for the null-hypothesis is not statistically valid; failing to reject the null hypothesis is not evidence for the null hypothesis. It's just as likely that the authors didn't use enough mice to have sufficient statistical power to show the effect. Indeed, visual inspection of Figure 3 makes this appear highly likely – clear changes in several activities can be seen in Figure 3 (most notably for % time moving, max acceleration and mean escape velocity). For instance, time moving declines from about 40% to 30%, and it's hard to believe that with additional samples this would remain non-significant, especially given that p = .0502. Relatedly, the authors claim that the correct critical value should be p = .0167 due to multiple comparisons. The meaning of a multiple-comparisons correction is difficult to interpret when the researcher's goal is to show that a difference isn't significant. Then, one could perform rank-sum tests on an arbitrarily large number of behavioral measures to reduce the critical value, and find no differences whatsoever between populations. A simple and more interpretable approach would be to simply report the effect sizes instead of p-values. Though there is a small reduction in exploratory behavior/ escape velocity, this is relatively small and essentially unavoidable due to the weight of any probe/casing being placed on the mouse's head.

We have re-written this section and amended our methods to no longer apply corrected p-value criteria. Ultimately, this does not change our claim because none of our comparisons achieved p<0.05, better yet the corrected p<0.0167 criteria. Text in the Results, Materials and methods, and Figure 3 legend has been changed to reflect this. Further, the overall language is toned down: the key conclusion is that implanted mice remain active and show species-typical behavior. Small and idiosyncratic changes in speed are in keeping with this conclusion. The new text in the Results section reads:

“To evaluate the suitability of the AMIE for use during behavior, we assessed the impact of the device on both spontaneous and stimulus-driven movements. […] Taken together, these behavioral observations argue that although the presence of the AMIE may have idiosyncratically slowed mice slightly, they remained active in an open arena and showed species-typical responses to threatening stimuli.”

We appreciate the reviewer’s recommendation to report effect sizes for these comparisons. A Cohen’s *d* for each comparison is also now reported on the figure.

3) Currently shipping probes do not have the same geometry as the probes described in this manuscript. That is, the probes that are now on sale at neuropixels.org have a slightly different shape than the "Phase 3A" probes employed in this paper. The authors need to describe how their system could be adapted to other probe geometries (and future designs will almost certainly evolve in this regard). It would be extremely valuable if the authors could provide an updated design based on the changes to the probes. Even though it may only be subtly different, it will save others from having to replicate the work.

We have modified our design to fit the Neuropixels 1.0 probes, and these designs are now available on our GitHub resource (https://github.com/churchlandlab/ChronicNeuropixels). We are working with collaborators at CSHL and UC San Diego to test these new designs with the Neuropixels 1.0 probes and will continue to update the available designs as needed, including making them available in Autodesk, which is freely available to students. We have also made corresponding notes in the text.

4) Reviewers had trouble visualizing the exact shape of the pieces and how they fit together. One suggestion was that the authors could make a short video that shows someone putting the pieces together by hand with a dummy probe. Some sort of additional visual aid such as this would greatly help readers to understand what screws onto what, how things slide, etc. Another suggestion was that it could perhaps somehow be done with the 3D renderings from Figure 1.

We agree and have generated a video (Video 1) which shows a 3D rendering.

5) A concern was raised about whether the authors' system for recovering the probe might compromise the stability of the probe, since most of the body of the probe itself is not really directly attached to the skull, but only via a thin 3D printed piece. A potential way to test this that was suggested could be to plot a comparison of waveforms like in Figure 4F, but for the same neuron recorded during periods of high acceleration versus stationary periods. An even better test would be to compute a metric of waveform similarity between moving and stationary periods, which could be compared to a recording in which the probe was attached in a fixed way. Perhaps such a recording would be available from Evans et al., who may have performed chronic recordings in freely moving animals with a probe fixed in place in the manner of Okun et al., 2016. Other reasonable ideas about how to address this concern would also be acceptable.

We agree that this is an important concern and have taken the reviewer’s suggestion to compare waveforms during moving versus stationary periods. For a session from Mouse

#7 (implanted with a recycled probe), we defined periods where the mouse was moving

(>15 cm/sec) or stationary (<5 cm/sec). A velocity of 15 cm/s is quite fast, while also common enough to enable the extraction of enough waveforms to compute a mean. We computed mean waveforms from 50 different spikes from 9 different units. As Author response image 1 shows, these waveforms are almost identical regardless of whether the mouse is moving or stationary. We computed correlation coefficients for moving vs. stationary mean waveforms, finding that all waveforms were very highly correlated (*r* > 0.96, *p* <.00001).

**Author response image 1. respfig1:** Comparison of mean waveforms (n=50) across periods where the mouse was moving (>15 cm/sec) or stationary (<5 cm/sec).

Several of these units show notable changes in absolute voltage, which could be for a number of reasons. For one, there are slow oscillations in the absolute voltage of the signal over time, so extracting a limited number of waveforms from different periods of the recording may reflect these slow changes in voltage. Second, it could be that the probe does slightly move relative to the recorded neuron during movement periods, leading to a change in the extracellularly recorded voltage. Additional analyses across multiple mice and probe depths would be needed to identify the cause of this shift. Regardless, it is clear that the waveforms recorded during movement and stationary periods are almost identical, providing further evidence that these recordings using our device are stable and that it is possible to isolate the same unit even in periods of fast movement. During manual spike sorting, we were also able to observe whether the waveforms changed over the course of the experiment, and found them to be quite stable.

Although we feel this was an important check, a full treatment of this topic warrants many additional analyses (e.g., multiple mice, different parameters to define movement periods) that are beyond the scope of this paper. We are however open to suggestions to include a reference in the manuscript to the preliminary analyses we have conducted if the reviewers or editors think this would be useful.

6) Some discussion of why this technology would be useful even if probes were less expensive and more accessible (or how/whether the approach would be modified under such conditions) would help extend the impact of this paper, assuming that limitations of the neuropixels purchasing scheme will be alleviated in the future.

We agree. In the revised Introduction, we highlight 3 advantages of our design. First, the AMIE can be readily adapted to different types of silicon probes, extending its significance beyond Neuropixels probes. Second, the AMIE is lighter than alternative designs because of the limited use of acrylic. Finally, by providing the drawings, materials and instructions, our device can be implemented not only by labs with years of expertise in electrophysiological recordings, but also by labs with different expertise that discover a new need to study neural activity during behavior. Existing methods for implanting electrodes are non-standardized, and rely on designs and surgical protocols that are part of lab lore and are not publicly available. This leaves new researchers in the dark about how to implant probes securely. Our thorough protocols make the AMIE within reach of these labs. Even if such labs are amongst those rare researchers without the need to re-use probes, they will still benefit from the use of standardized, open source hardware that this manuscript provides.